

# Integrated bulk and single-cell RNA sequencing identifies an aneuploidy-based gene signature to predict sensitivity of lung adenocarcinoma to traditional chemotherapy drugs and patients' prognosis

Xiaobin Wang[1,*], Jiakuan Chen[1,*], Chaofan Li[2], Yufei Liu[2], Shiqun Chen[3], Feng Lv[1], Ke Lan[1], Wei He[2], Hongsheng Zhu[4], Liang Xu[4], Kaiyuan Ma[4] and Haihua Guo[1]

[1] Department of Thoracic Surgery, Tangdu Hospital, Air Force Military Medical University, Xi'an, China
[2] Department of Thoracic Surgery, The 986 Military Medical Hospital of the Air Force, Xi'an, China
[3] Thoracic Surgery, Weinan Central Hospital, Weinan, China
[4] Thoracic Surgery, Shaanxi Chenggu County Hospital, Chenggu, China
* These authors contributed equally to this work.

Corresponding author
Haihua Guo, guohh369@163.com

## ABSTRACT

**Background:** Patients with lung adenocarcinoma (LUAD) often develop a poor prognosis. Currently, researches on prognostic and immunotherapeutic capacity of aneuploidy-related genes in LUAD are limited.

**Methods:** Genes related to aneuploidy were screened based on bulk RNA sequencing data from public databases using Spearman method. Next, univariate Cox and Lasso regression analyses were performed to establish an aneuploidy-related riskscore (ARS) model. Results derived from bioinformatics analysis were further validated using cellular experiments. In addition, typical LUAD cells were identified by subtype clustering, followed by SCENIC and intercellular communication analyses. Finally, ESTIMATE, ssGSEA and CIBERSORT algorithms were employed to analyze the potential relationship between ARS and tumor immune environment.

**Results:** A five-gene ARS signature was developed. These genes were abnormally high-expressed in LUAD cell lines, and in particular the high expression of CKS1B promoted the proliferative, migratory and invasive phenotypes of LUAD cell lines. Low ARS group had longer overall survival time, higher degrees of inflammatory infiltration, and could benefit more from receiving immunotherapy. Patients in low ASR group responded more actively to traditional chemotherapy drugs (Erlotinib and Roscovitine). The scRNA-seq analysis annotated 17 cell subpopulations into seven cell clusters. Core transcription factors (TFs) such as CREB3L1 and CEBPD were enriched in high ARS cell group, while TFs such as BCLAF1 and UQCRB were enriched in low ARS cell group. CellChat analysis revealed that high ARS cell groups communicated with immune cells *via* SPP1 (ITGA4-ITGB1) and MK (MDK-NCl) signaling pathways.

**Conclusion:** In this research, integrative analysis based on the ARS model provided a potential direction for improving the diagnosis and treatment of LUAD.

## INTRODUCTION

Lung adenocarcinoma (LUAD), as the main type of lung cancer (*Sung et al., 2021*; *Schabath & Cote, 2019*), has become a major cause of cancer-related mortality globally (*Siegel et al., 2023*). The strong proliferation and invasion ability of LUAD cells is closely related to the poor survival rate of the cancer (*Siegel, Miller & Jemal, 2019*; *Liu et al., 2021*). Currently, chemotherapy remains still one of the primary therapy for LUAD but its efficacy is greatly restricted by patients' resistance (*Zhang et al., 2020*). For this reason, from 2015, the World Health Organization has encouraged to identify molecular markers to help diagnose LUAD (*Nicholson et al., 2022*; *Thai et al., 2021*).

Aneuploidy is a form of abnormality in the number of chromosomes. As massive genetic mutation, aneuploidy enables cells to adjust to changes in internal and external environment, and facilitates tumor occurrence by promoting genetic instability, diversity, and evolution (*Ben-David & Amon, 2020*). Around half of blood cancers and 90% of solid tumors exhibit different forms of aneuploidy, which has been found to be associated with tumor progression and unfavorable prognosis (*Xian et al., 2021*; *Stopsack et al., 2019*). Aneuploidy triggers stress response in tumor cells, initiating or promoting the imbalance of local immune cells to increase the risk of immune escape (*Giam & Rancati, 2015*). Based on aneuploidy-related genes, *Liu et al. (2023b)* identified three molecular subtypes of head and neck squamous cell carcinomas and developed a risk model predictive of the prognosis of the cancer. *Jia et al. (2021)* discovered that a higher aneuploidy score was related to more limited clinical benefits from radiotherapy in non-small cell lung cancer (*Yang et al., 2021*). *Taylor et al. (2018)* disclosed loss of chromosome 3p and gains of chromosome arm1q and 3q using lung cancer dataset. However, using aneuploidy-related gene markers to facilitate the diagnosis or treatment of LUAD have been rarely studied.

High-throughput transcriptomics enabled by bulk RNA determines average gene expression profiles of all the cells in a cluster, and signal-cell RNA sequencing (scRNA-seq) clarifies the heterogeneity of cell types (*Kodous, Balaiah & Ramanathan, 2023*). Recently, scRNA-seq research has been widely applied in many research fields including in the studies of tumor ecosystems (*Yan et al., 2021*; *Hornburg et al., 2021*), immune-related diseases (*Lin et al., 2022*; *Wang et al., 2021*) and neuroscience (*Galiakberova et al., 2023*). Analysis on the variations and regulatory mechanism of cell subpopulation specificity at single-cell level can help better understand the pathological and physiological processes underlying LUAD development.
The use of scRNA-seq technology contributes to the development of cancer diagnosis, personalized therapeutic strategies as well as how to combat treatment resistance (*Kinker et al., 2020*). Thus, in this work, we systematically explored the intra-cancer cell heterogeneity of LUAD samples by integrating bulk and scRNA-seq data. An independent prognostic model was developed based on the foundation of aneuploidy-related prognostic genes from bulk transcriptome data of LUAD. The prognostic significance of the signature was confirmed using separate queues. The regulation of malignant phenotype of LUAD by the model genes was confirmed with cellular experiments. Immunotherapy for different LUAD patients and sensitive traditional chemotherapy drugs were analyzed. A total of seven cell clusters were annotated using scRNA-seq data, and five of them were immunocytes. Based on the aneuploidy-related riskscore (ARS) model, tumor subpopulations were stratified into high and low ARS cell groups. Regulons based on TF-target and communication signaling pathways between cells differed between the two cell groups. In short, the current discoveries provided a novel direction for the diagnosis and treatment of LUAD.

## MATERIALS AND METHODS

### Study source and data preprocessing

A pan-cancer genomic aneuploidy score was downloaded from a previous study (*Taylor et al., 2018*). After excluding the aneuploidy score of other cancer samples, The Cancer Genome Atlas Lung Adenocarcinoma (TCGA-LUAD) samples with aneuploidy score were retained for further study (Table S1). Subsequently, the RNA-Seq data of TCGA-LUAD were acquired from the The Cancer Genoma Atlas GDC application programming interface (API) website. For validation, GSE31210 (*Okayama et al., 2012*) and GSE30219 (*Rousseaux et al., 2013*) data were downloaded from the GEO database. For TCGA-LUAD samples, samples with clinical information and survival time >0 days were retained but those without survival status were removed. Finally, a total of 483 LUAD samples were compiled for further analysis (Table S2). For GEO data, samples without survival status were deleted, and mean expression value of the genes was adopted when multiple probes matched to one gene. All the data used in this article were derived from publicly available databases without access restriction and therefore did not require additional ethical approval. Data collection and analysis were performed strictly following the relevant rules and regulations.

### Selection of differentially expressed genes (DEGs) related to aneuploidy and functional annotation

Differential gene analysis between tumor and adjacent tissue samples was conducted using the TCGA-LUAD database. False discovery rate (FDR) <0.05 and |log2FC| > 1 were the screening criteria. At the same time, the correlation between DEGs and aneuploidy was analyzed using Spearman method (R > 0.3 and $p < 0.05$). Finally, the clusterprofiler package (*Yu et al., 2012*) was employed to perform Gene Ontology (GO) and Kyoto Encyclopedia of Genes and Genomes (KEGG) enrichment analysis on these aneuploidy-related DEGs.

### Developing a riskscore model based on aneuploidy-related genes

The TCGA dataset and the GEO dataset served as a training and a validation set, respectively. Prognostic genes for LUAD were screened from all the aneuploidy-related DEGs using univariate Cox analysis, followed by performing Lasso regression to shrink the number of genes in the risk model. The risk score for each patient was calculated using the following Eq. (1):

$$ARS = \sum \beta i \times Expi \tag{1}$$

where Expi represented the expression of prognosis-related gene, β represented the Cox regression coefficient of the corresponding gene.

Samples with ARS score > 0 after zscore transformation were assigned into high-ARS group, while those with ARS score < 0 were assigned into low-ARS group. Survival curves were plotted using Kaplan-Meier for prognosis analysis. Additionally, the log-rank test was used to assess the significance of the differences.

### Immune landscape analysis

To comprehensively assess the immune cell composition and activity in LUAD, we used two complementary computational approaches (ssGSEA and CIBERSORT). Based on the reference dataset from *Charoentong et al.*'s *(2017)* study ssGSEA was used to calculate the score of 28 immune cells in each sample. In addition, CIBERSORT is a more precise quantification method that estimates the proportion of 22 immune cells using a predefined set of immune cell-specific marker genes (*Chen et al., 2018*).

### Immunotherapy and drug sensitivity evaluation

An immunotherapy cohort (IMvigor210) was included to evaluate the influence of ARS on predicting immunotherapy sensitivity. The IMvigor210 cohort of patients demonstrated different degrees of responses to PD-L1 receptor blockers, including progressive disease (PD), stable disease (SD), partial response (PR), and complete response (CR). In addition, pRRophic package (*Geeleher, Cox & Huang, 2014*) was adopted to estimate potential sensitive chemotherapy drugs between two ARS groups in the TCGA queue. A lower half maximal inhibitory concentration (IC50) value indicated a higher sensitivity to drugs.

### Single-cell RNA sequencing analysis of LUAD samples

The GSE203360 dataset (*He et al., 2022*) including six LUAD samples was used for scRNA sequencing analysis. The R package Seurat (*Stuart et al., 2019*) (v4.0.4) was adopted for following analysis. Cells were primarily filtered based on the criteria that each gene expressed at least in three cells and each cell contained more than 200 expressed genes. Cells with mitochondrial genes >25% were excluded to delete aging cells with limited biological significance. Subsequently, dimensionality reduction clustering analysis was performed employing principal component analysis (PCA), t-distributed stochastic neighbor embedding (t-SNE), FindNeighbor and FindCluster function. Based on cell markers from the CellMarker2.0 website, initial subpopulations were reclassified according

to high-expressed marker genes. The DEGs in annotated subgroups were screened by FindAllMarkers function, followed by KEGG enrichment analysis.

## Screening of differential regulons and cell communications in tumor cells divided by ARS

Cancer cells were extracted from scRNA-seq data. The ARS score of each cell was calculated using the Eq. (2). Thereafter, these cancer cells were stratified into high and low ARS cell groups under the same grouping criteria used in TCGA-LUAD queue. The human RcisTarget database was downloaded from website (https://resources.aertslab. org/cistarget/). R package "SCENIC" (*Van de Sande et al., 2020*) was employed for the development of TF regulatory networks. A potential TF-target regulatory relationship was established first and then motif analysis was used to eliminate genes indirectly regulated by TF. A regulon is a set of TFs and their directly regulated target genes. The connections among the regulons were analyzed using connection specific indicators (CSI). CellChat software (*Jin et al., 2021*), a free online R package (https://GitHub.com/sqjin/CellChat), was applied to infer intercellular communications and select predominant receptor–ligand pairs.

## Cell culture and transient transfection

Human LUAD cell lines including H1299, A549 and human normal bronchial epithelial cell BEAS-2B were commercially obtained from BNCC (Beijing, China). DMEM F12 with 10% FBS (Gibco, Thermo Fisher, Massachusetts, USA) was used for culturing A549 and BEAS-2B cells in a humidified environment at 37 °C with 5% $CO_2$. The H1299, A549 cell lines were transfected with CKS1B siRNA (Dharmacon, Lafayette, CO, USA) utilizing Lipofectamine RNAi Max (Invitrogen, Waltham, MA, USA). The target sequence for CKS1B siRNA was CGCACAAACAAATTTACTATTCG (5′–3′).

## QRT-PCR

Total RNA was separated using TRIzol (Thermo Fisher, Waltham, MA, USA) reagent. QRT-PCR was conducted on the extracted RNA from each sample (2 μg) using FastStart Universal SYBR Green Master on a LightCycler 480 PCR System (Roche, Switzerland, USA). The cDNA consisted of a total reaction volume of 20 μl (an appropriate water volume, 0.5 μl of forward and reverse primers, 2 μl of cDNA template, and 10 μl of PCR mixture) and served as a template. PCR reaction was reacted following the cycling conditions that began with initial DNA denaturation phase at 95 °C for 30 s, followed by 45 cycles at 94 °C for 15 s, at 56 °C for 30 s, and at 72 °C for 20 s. Each sample was run in triplicate. Threshold cycle (CT) data were obtained and standardized to the level of GAPDH applying the $2^{-\Delta\Delta CT}$ method. The mRNA expression in both tumor and normal control tissues was compared. The sequences of primer pairs for the genes were listed in Table 1.

## Transwell assay

Migration and invasion of H1299 and A549 cell lines were detected by Transwell assays. Briefly, chambers coated (for invasion) or uncoated with Matrigel (BD Biosciences,

**Table 1 The sequences of primer pairs for the target genes of LUAD.**

| Gene | Forward primer sequence (5′-3′) | Reverse primer sequence (5′-3′) |
|------|--------------------------------|--------------------------------|
| PLK1 | GCACAGTGTCAATGCCTCCAAG | GCCGTACTTGTCCGAATAGTCC |
| ANLN | CAGACAGTTCCATCCAAGGGAG | CTTGACAACGCTCTCCAAAGCG |
| HMMR | GGCTGGGAAAAATGCAGAGGATG | CCTTTAGTGCTGACTTGGTCTGC |
| CKS1B | GGAATCTTGGCGTTCAGCAGAG | GAGGCTGAAAAGTAGCTTGCCAG |
| UBE2S | CGATGGCATCAAGGTCTTTCCC | CAGCAGGAGTTTCATGCGGAAC |
| GAPDH | GTCTCCTCTGACTTCAACAGCG | ACCACCCTGTTGCTGTAGCCAA |

Franklin Lakes, NJ, USA) (for migration) were first inoculated with cells ($5 \times 10^4$). Serum-free medium and complete DMEM medium were added to the upper and lower layer, respectively. After incubation for 24 h, 4% paraformaldehyde was used for fixing migrating or invading cells, followed by cell dyeing using crystalline violet (0.1%).

## Cell viability

Cell viability was detected by performing cell counting kit-8 assay (Beyotime, Beijing, China). Specifically, the cells were different treatments were cultured in 96-well plates at a density of $1 \times 10^3$ cells/well and added with CCK-8 solution at indicated time points. After 2-h incubation at 37 °C, a microplate reader (Thermo Fisher, Waltham, MA, USA) was utilized to measure the O.D 450 values of each well.

## Statistical analysis

All analysis were conducted utilizing R package software (version 4). Survival data were summarized using Kaplan–Meier approach, and statistical significance was detected by the log-rank test. The time receiver operating characteristic (ROC) curves were adopted to evaluate the prediction ability of the model. Student t-tests were used to determine statistical significance with a one-way ANOVA used for comparative testing. A $p < 0.05$ was thought to be statistically significant. SangerBox (http://sangerbox.com/) provided analytical assistance (*Shen et al., 2022*).

# RESULTS

## DEGs related to aneuploidy and functional annotation

Firstly, RNA-seq data of the TCGA-LUAD dataset were used to screen differential genes (FDR < 0.05 and |log2FC| > 1) between tumor and para-cancerous tissue samples and a total of 3,316 DEGs were obtained for subsequent analysis (Fig. 1A). Correlation analysis detected a significant correlation between 182 DEGs and AS, and these 182 DEGs were high-expressed in tumor tissue (Fig. 1B). Further gene functional enrichment analysis showed that these 182 DEGs were enriched mainly in DNA replication and cell cycle-related pathways (Figs. 1C–1F).

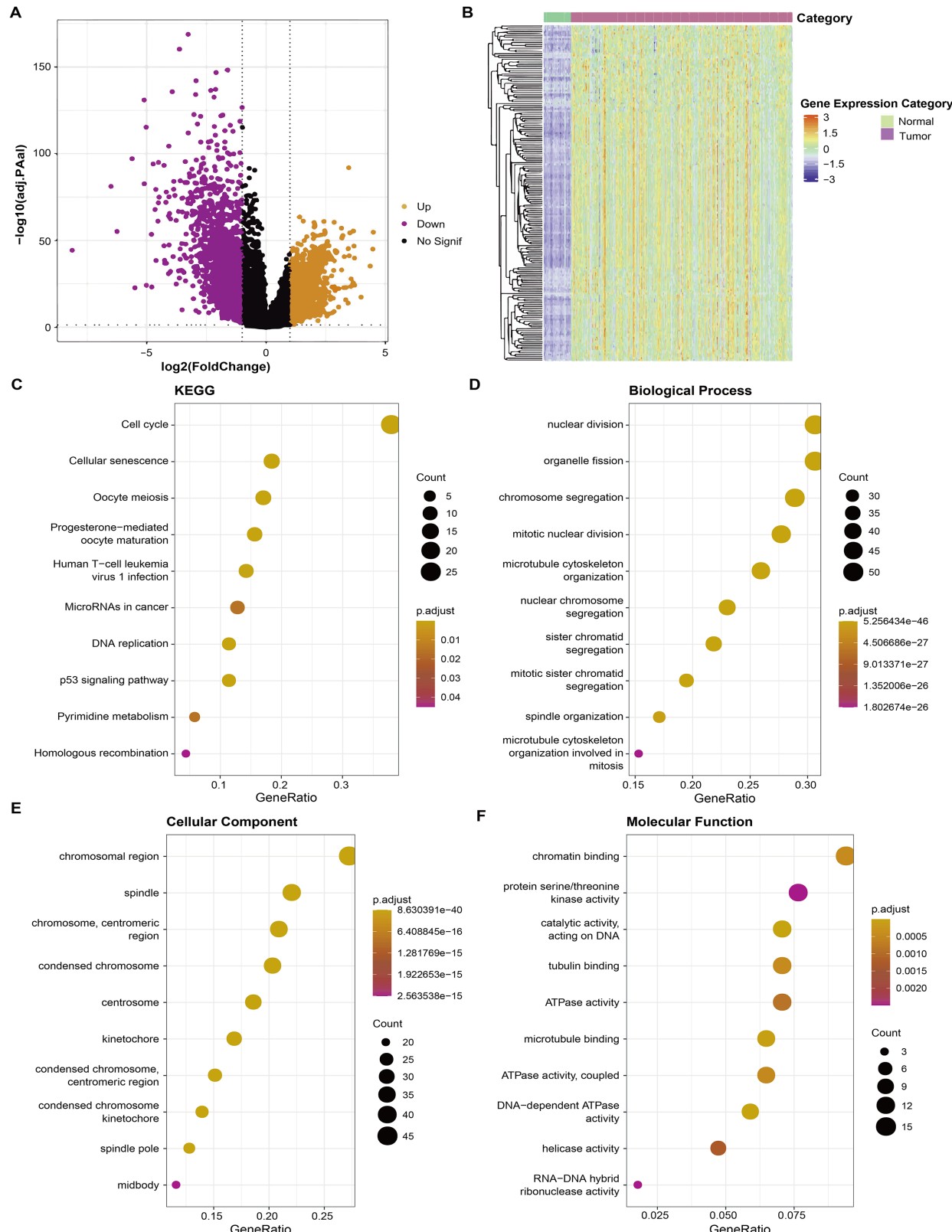

**Figure 1 Identification of aneuploidy relevant DEGs and functional annotation.** (A) DGEs between tumor and normal tissues in the TCGA cohort. (B) The heatmap depicting the expression levels of DEGs related to aneuploidy in tumor and normal tissues. (C) KEGG analysis of DEGs related to aneuploidy (D–F) GO analysis of DEGs related to aneuploidy.

## Establishment and assessment of the riskscore model based on the aneuploidy-related genes

Univariate Cox analysis identified 125 genes with significant prognostic impact ($p < 0.01$) (Table S3). Lasso cox regression in R software package glmnet (*Simon et al., 2011*) was used to reduce the gene number. Ten-fold cross validation was applied to develop the model. The model was the optimal when penalty parameter lambda was 0.038 (Figs. 2A and 2B), therefore, five genes at lambda = 0.038 were chosen as key prognostic genes. The coefficient of the corresponding gene was shown in Fig. 2C. Each patient's riskscore was calculated following Eq. (2).

$$ARS = -0.004 \times \text{Exp PLK1} + 0.208 \times \text{Exp ANLN} + 0.059 \times \text{Exp HMMR} + 0.033 \times \text{Exp CKS1B} + 0.073 \times \text{Exp UBE2S} \tag{2}$$

Based on Eq. (2), samples in TCGA-LUAD queue were separated into high and low ARS groups. It could be observed that the OS time in low ARS group was longer than that in high-ARS group ($p < 0.0001$, Fig. 2D). We observed that in the TCGA cohort, the area under the ROC curve (AUC) was 0.68, 0.65, and 0.67 for 1-, 3-, and 5-year OS, respectively (Fig. 2D). The five key genes were all overexpressed in tumor tissues (Fig. 2D). The model was confirmed using GSE31210 and GSE30219 datasets (Figs. 2E and 2F). Specifically, patients in the GSE31210 cohort had AUC values of 0.63, 0.68, and 0.74 for 1-, 3-, and 5-year OS, respectively, while those in the GSE30219 cohort had AUC values of 0.77, 0.74, and 0.75 for 1-, 3-, and 5-year OS, respectively.

## The ARS model was an independent indicator for LUAD prognosis

To verify the applicability of the five-gene ARS signature in clinical setting, COX regression was performed. In the TCGA cohort, staging and risk scores and patients' OS were significantly correlated ($p < 0.001$) (Fig. S1A). Multivariate analysis showed that staging and risk score were also independent predictors (Fig. S1B). Subsequently, similar results were found in the GSE30219 cohort, and age, staging and risk score were also the independent predictors of LUAD prognosis ($p < 0.001$) (Figs. S1D, S1E). Importantly, the nomogram all showed the same predictive function (Figs. S1C, S1F). These results suggested that the risk model developed based on these two datasets was a favorable tool for predicting the prognostic outcomes of LUAD patients, and that both staging and risk scores were independent prognostic factors.

## CKS1B was a characteristic gene in the ARS to promote the malignant phenotypes of LUAD

To validate the reliability of the ARS model in predicting the prognosis of LUAD and the correlation between the model genes and LUAD progression, we measured the expression of PLK1, ANLN, HMMR, CKS1B and UBE2S in LUAD cell lines H1299 and A549 and bronchial normal epithelial cells BEAS-2B using PCR. Consistent with the bioinformatics results, the level of PLK1, ANLN, HMMR, CKS1B and UBE2S was elevated in the H1299 and A549 cell lines, in particular CKS1B was significantly upregulated in tumor cell lines (Figs. 3A–3E). Next, changes in migration and invasive of H1299 and A549 after inhibiting

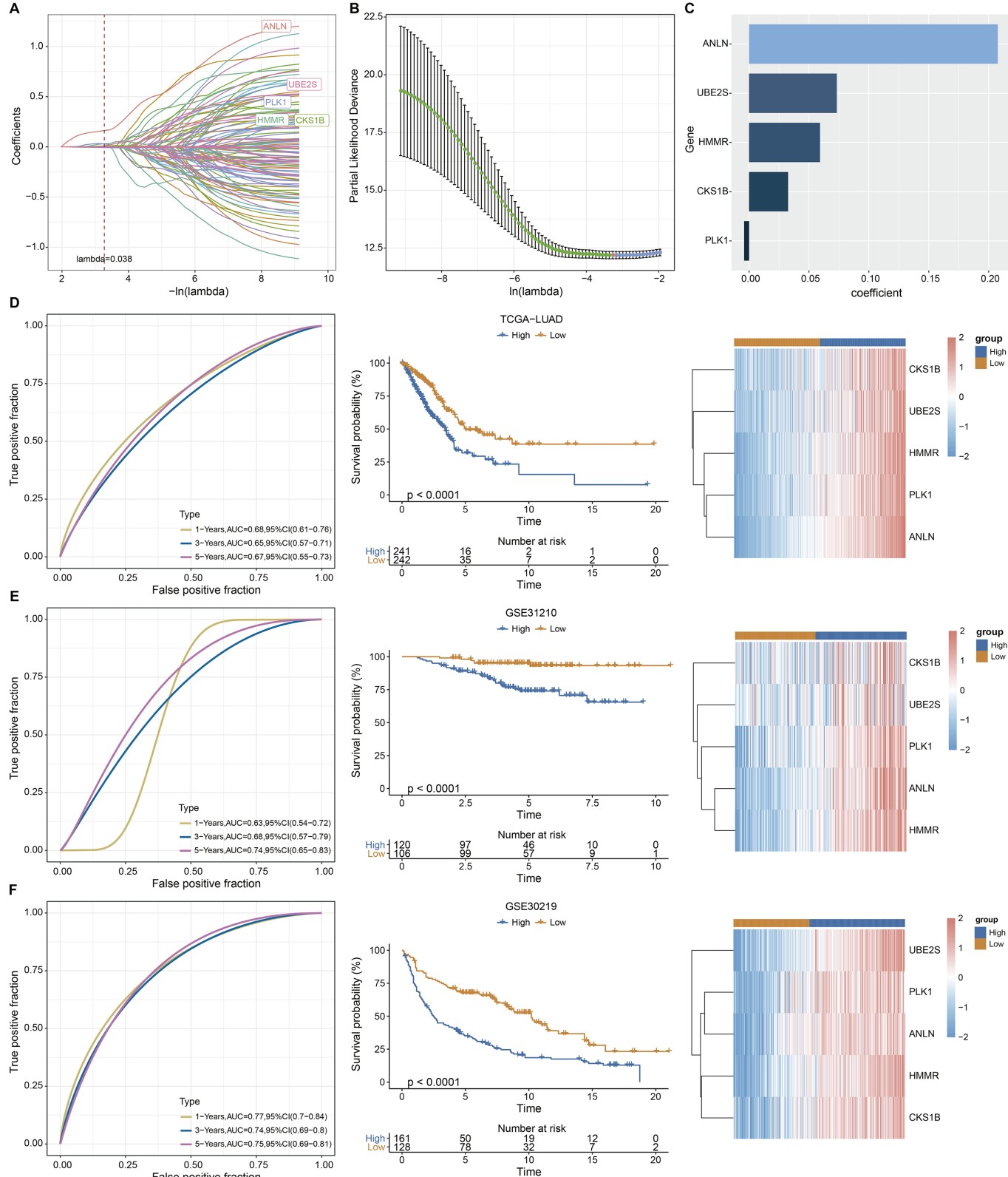

**Figure 2 Construction and evaluation of the ARS model based on aneuploidy related genes.** (A and B) Lasso regression analysis. (C) Multi-factor coefficient of five hub genes. (D) Time-ROC, survival curves and gene expression analysis in TCGA testing cohort. (E–F) Time-ROC, survival curves and gene expression analysis in GSE31210 and GSE30219 validating cohorts.

the expression of CKS1B in the two cell lines were examined. We found that CKS1B expression promoted the development of LUAD, and that the migration and invasion abilities of the two cell lines were significantly suppressed after inhibition of CKS1B (Figs. 3F and 3G). In addition, silencing CKS1B expression affected the viability of H1299 and A549 cell lines relative to controls (Figs. 3H and 3I). The above results demonstrated the reliability of using ARS model to predict the prognostic survival of LUAD, and in particular, the model genes significantly contributed to the malignant phenotype of LUAD.

## Analysis of immune cell infiltration in the two cohorts

As shown in Fig. 4A, most of immune cells including T follicular helper cell, central memory CD4 T cell, immature B cell, activated B cell, immature, activated and plasmacytoid dendritic cell, eosinophil cell, macrophage, type 17 T helper cell, mast cell, monocyte and natural killer cell had higher score in low ARS group ($p < 0.05$), while few immunocytes such as activated, effector memory CD4 T cell, memory B cell, Type 2 T helper cell, gamma delta T cell, natural killer T cell and neutrophil were enriched in high ARS group. CIBERSORT was used to perform a more detailed assessment on different states of the immune cells. It could be found that low ARS group had obviously a higher proportion of T cells CD4 memory resting, Tregs, dendritic cells resting, and mast cells resting ($p < 0.01$), whereas that of T cells CD4 memory activated and macrophages at M0 and M1 stage were noticeably reduced ($p < 0.0001$, Fig. 4B).

## The responsiveness of ARS model to immunotherapy and drug sensibility

Immunotherapy is a treatment that fights against cancer with synergistic survival benefits (*Curran et al., 2010*). Herein, we tested the prognostic significance of ARS model in immunotherapy dataset IMvigor210 cohort. Kaplan-Meier curves displayed that patient with a higher ARS had shorter OS ($p < 0.05$, Fig. 5A). Among them, SD/PD patients exhibited a higher ARS than CR/PR patients ($p < 0.05$, Fig. 5B). In high ARS group, the proportion of SD/PD was higher than that in the low ARS group (Fig. 5C). Applying pRRophic package, we further predicted potential sensitive drugs for different patients and found that patients in high ARS group were more sensitive to cosplatin, BI-2536, pyrimethamine and GW843682X, whereas those in low ARS group were more responsive to erlotinib and roscovitine (Figs. 5D and 5E).

## Single-cell landscape of LUAD samples

*He et al. (2022)* generated scRNA-seq count matrix. PCA and t-SNE dimension reduction were performed based on six specimens derived from invasive adenocarcinoma patients. A total of 17 subpopulations were obtained applying FindNeighbor and FindCluster (Dim = 30, Resolution = 0.5) and reclassified according to the expression of cell markers in the subpopulations (Fig. S2). Finally, seven cell types were determined and denoted as alveolar epithelial cell, B cells, cancer cells, mast cells, monocytes/macrophages, myeloid dendritic cells and T cells (Figs. 6A and 6B). By using the FindAllMarkers function under the screening criteria of logfc = 0.5 (multiple of differences) and Minpct = 0.25 (smallest

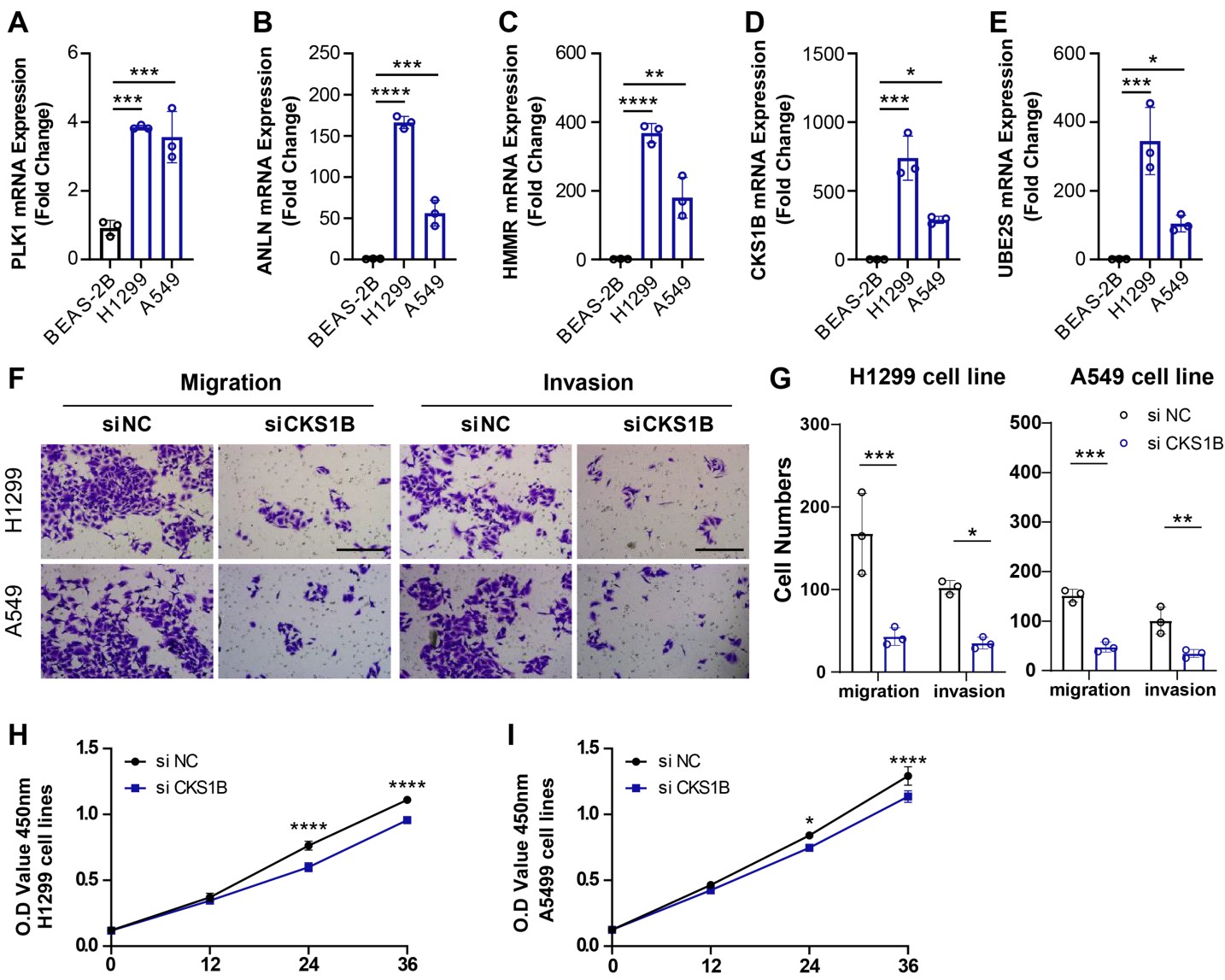

**Figure 3 Role of ARS characterized genes on the malignant phenotype of LUAD cell lines.** (A–E) PCR was performed to detect the expression of PLK1, ANLN, HMMR, CKS1B, and UBE2S in BEAS-2B, H1299, and A549 cell lines, and relative quantification was performed ($n$ = 3). (F and G) Alterations in cell migration and invasive capacity following inhibition of CKS1B expression in H1299 and A549 cell lines and quantification of cell numbers ($n$ = 3). (H and I) Changes in cell viability from 0–36 h after inhibition of CKS1B expression in H1299 and A549 cell lines and quantification of O.D Value ($n$ = 3). * ≤ 0.05, ** ≤ 0.01, *** ≤ 0.001, **** ≤ 0.0001. $N$ = 3, The results are presented as mean ± SD.

expression ratio of differential genes), the DEGs in the seven cell types were retained and the top five genes were shown in a heatmap (Fig. 6C). Subsequently, to explore the function of these cell subpopulations, we selected the top five genes for enrichment analysis. Figure 6D showed the enrichment of marker genes in different pathways in different cell types. Among them, monocytes/macrophages were mainly involved in pathways such as phagosome and lysosome. B and T cells were also predominantly enriched in the pathway of protein processing in endoplasmic reticulum. These results may provide insight into the

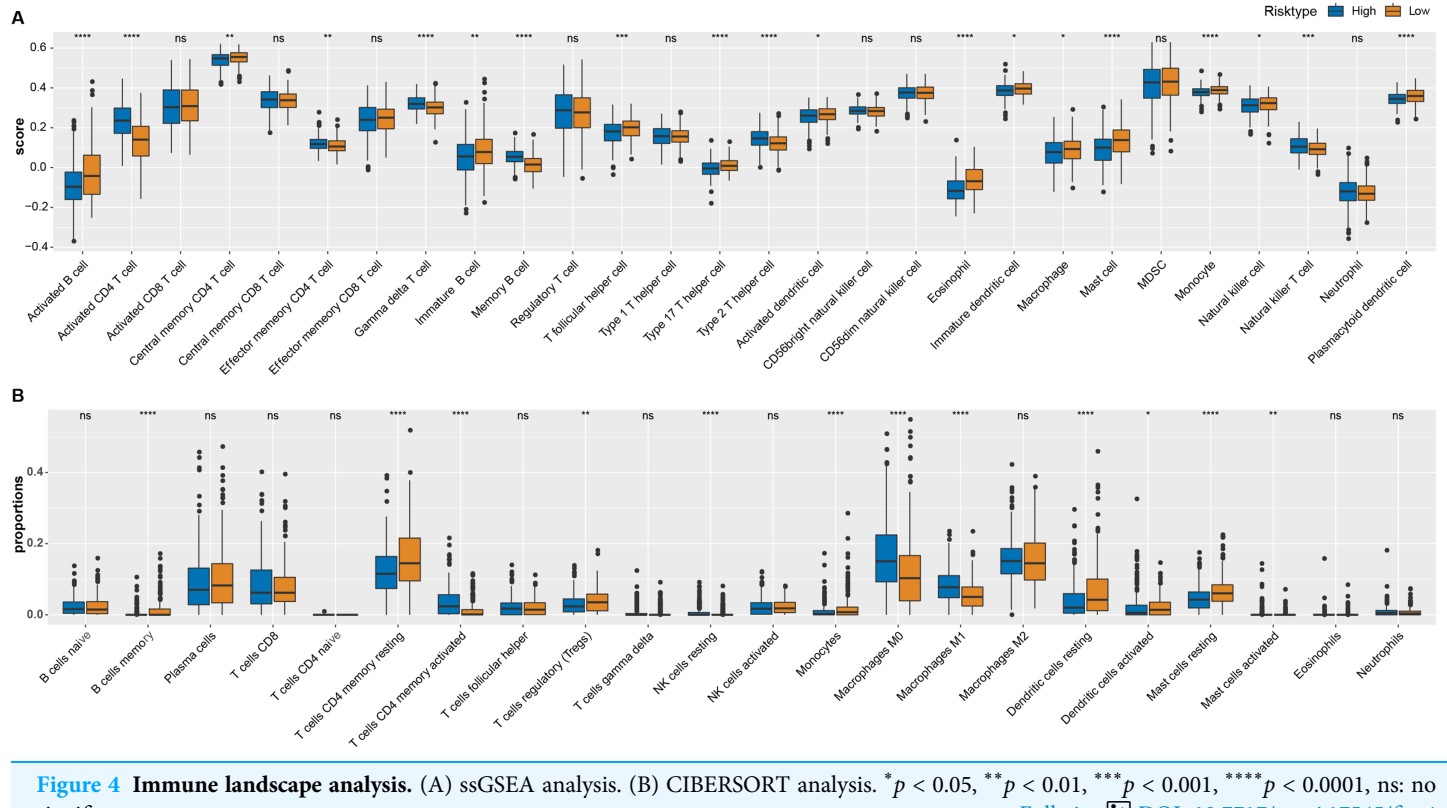

**Figure 4 Immune landscape analysis.** (A) ssGSEA analysis. (B) CIBERSORT analysis. $^*p < 0.05$, $^{**}p < 0.01$, $^{***}p < 0.001$, $^{****}p < 0.0001$, ns: no significance.

complex biology within LUAD and offer potential targets for further improve personalized treatment for LUAD patients.

## TF-target connection and function analysis

Applying SCENIC algorithm and CSI, key regulons were identified and clustered. As seen in Fig. 7A, regulons with similar CSI were clustered together. Regulons with analogous high CSI may jointly regulate downstream genes and were involved in cell functions. Next, we calculated the regulon specificity score (RSS) for the two ARS cell groups. The horizontal and vertical axes represented the ranking of the regulons and the RSS, respectively. The top five TFs in the ranking were represented by red dots. A higher RSS indicated a closer relationship between the regulons and cell type specificity. As shown in Figs. 7B and 7C, regulons such as CREB3L1, JUN_extended, CEBPD, FOS_extended and JUNB_extended were enriched in high ARS cell group. Nevertheless, the regulons including BCLAF1_extended, TAF7_extended, RAD21_extended, BCLAF1 and UQCRB ranked the top 5 in low ARS cell group.

## CellChat identified communication patterns between immune cells and LUAD cells

Cell communication analysis helps to analyze the interaction between cells, explore the tumor immune microenvironment and dig potential therapeutic targets for diseases (*Jin et al., 2021*). As mentioned above, high and low ARS groups had different infiltration status

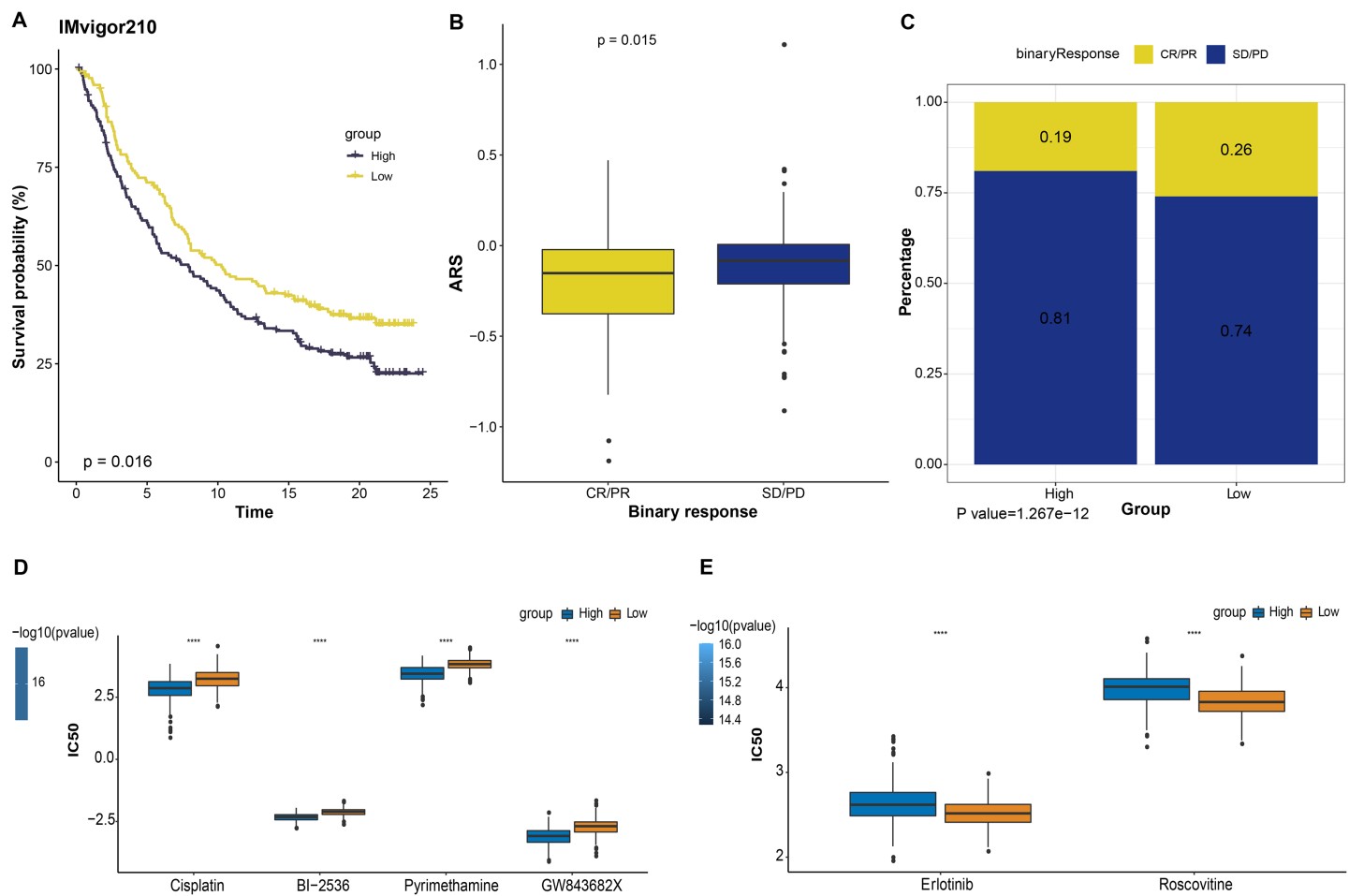

**Figure 5 The responsiveness of ARS model to immunotherapy and drug sensibility.** (A) Kaplan-Meier curves analysis in IMvigor210 immunotherapy database based on ARS model. (B) ARS difference in immunotherapy responses. (C) Immunotherapy difference between high and low ARS groups. (D) Drugs sensitive to high ARS group. (E) Drugs sensitive to low ARS group. ****$p < 0.0001$.

of immune cells. Herein, we investigated the communication between immune cells and LUAD cells using scRNA-seq analysis. The results demonstrated that the cancer cells in high ARS group could receive signals from immune cells through the SPP1 signaling pathway *via* ligand receptor pair ITGA4-ITGB1 (Fig. 8A). As shown in Fig. 8B, tumor cells communicated with other immune cells through the MK signaling pathway *via* ligand receptor pair MDK-NCl. Moreover, high ARS group had thicker lines, which indicated stronger interaction signals. Communications through GDF (GDF15-TGFBR2) and ECF (AREG-EGRF) signaling pathway between high and low ARS cell groups were also found (Figs. 8C and 8D). These results demonstrated that ARS helped to detect an enhanced tumor-immune system interplay in LUAD.

## DISCUSSION

LUAD is represented by highly inflammatory infiltration (*Luo et al., 2020*; *Zheng et al., 2023*). The early asymptomatic characteristics of LUAD often leads to a late diagnosis of

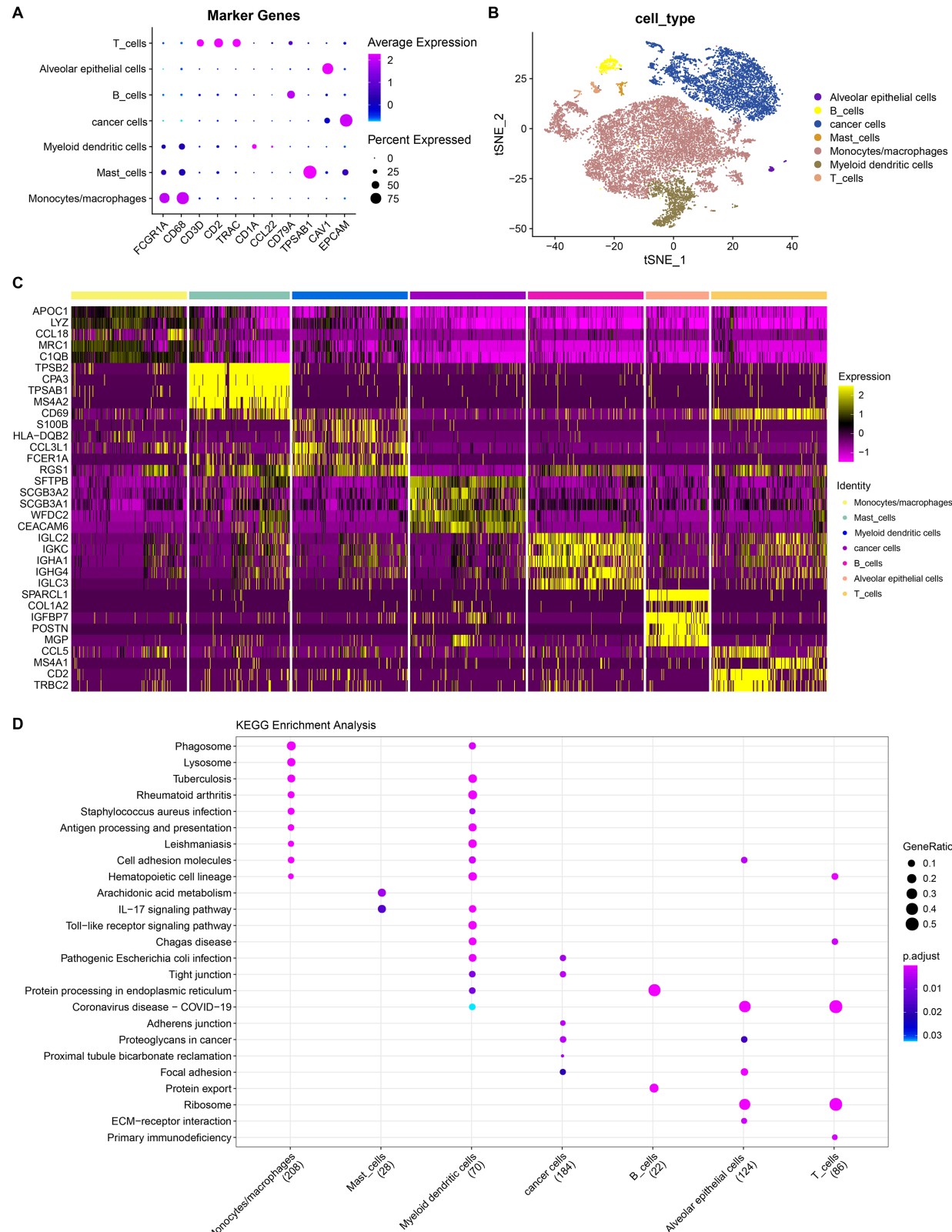

**Figure 6 Landscape plots of LUAD scRNA-seq data.** (A) Cell marker genes of different cell subpopulations. (B) Annotated cell subpopulations distribution in t-SNE diagram. (C) A heatmap of the top 5 DEGs in each cell subpopulation. (D) KEGG enrichment analysis of top five DEGs in each cell subpopulations.
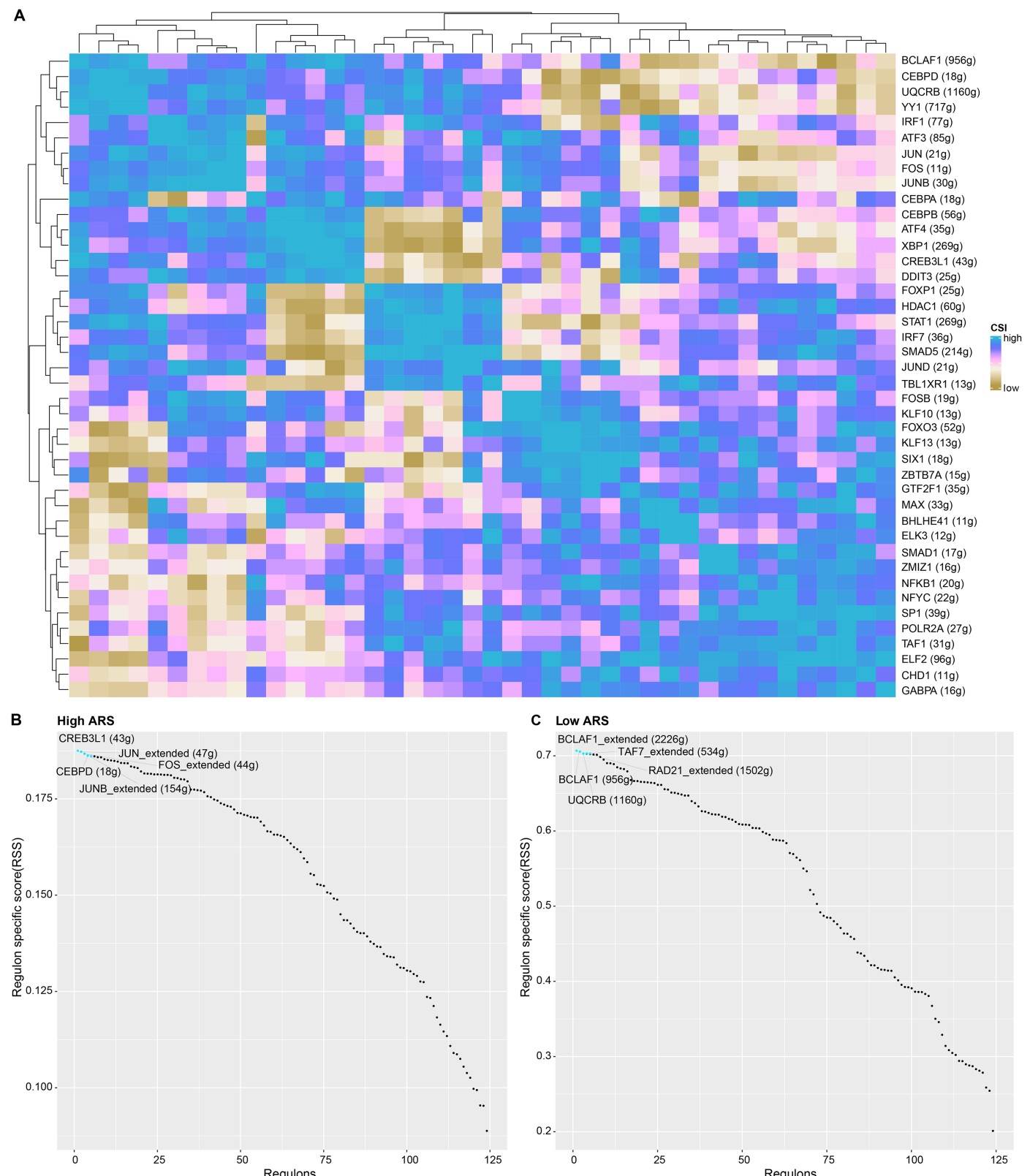

**Figure 7 The analysis of regulons in high and low ARS cell groups.** (A) A cluster graph based on regulon's CSI (the rows and columns both represented regulons). (B) Regulons ranked top five in high ARS cell group. (C) Regulons ranked top five in low ARS cell group.

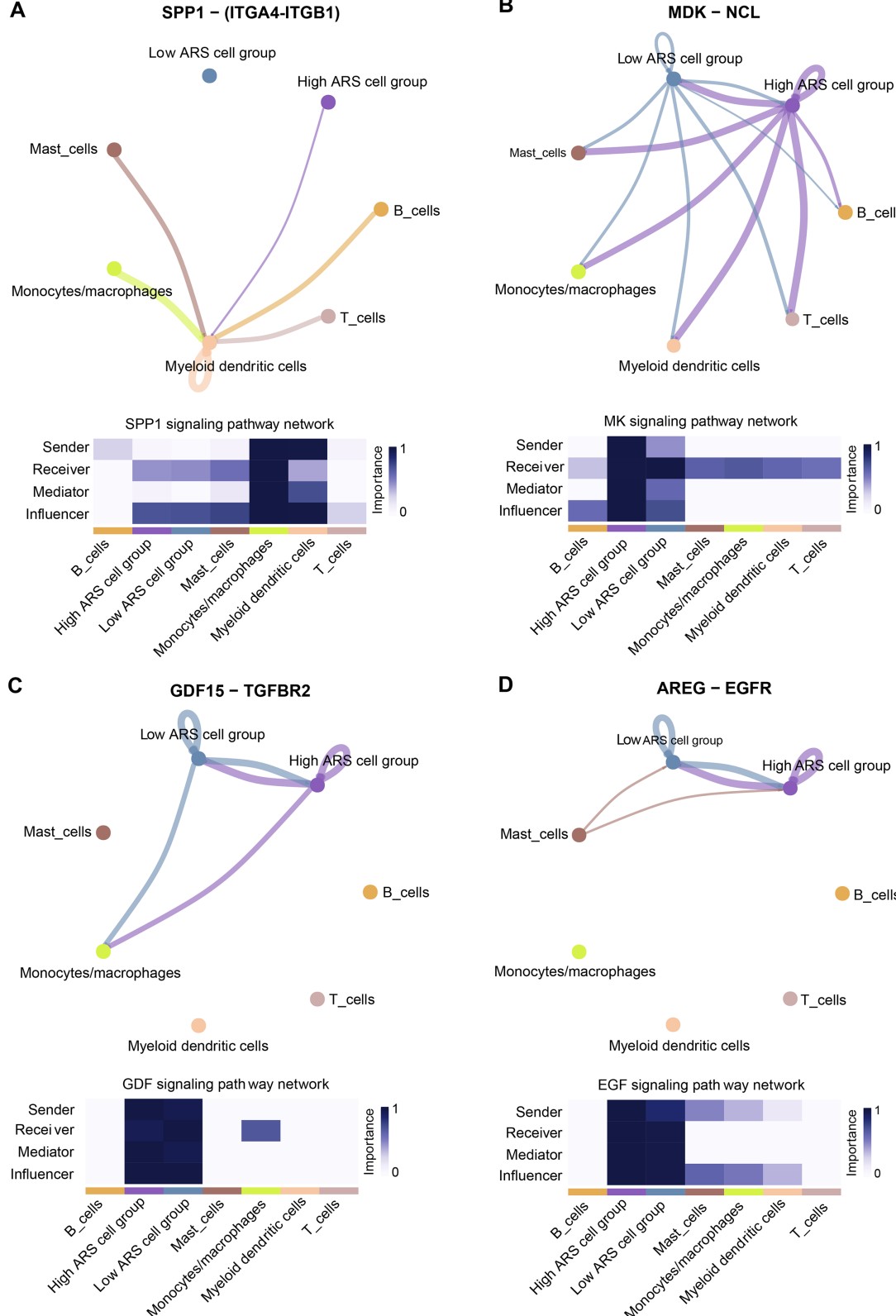

**Figure 8 Overview of the CellChat communications between cancer and immune cells.** (A) High ARS cell group received signals from immune cells *via* SPP1-(ITGA4-ITGB1) pathway. (B) Cancer cells sent signals to immune cells *via* the MK-(MDK-NCL) pathway. (C and D) High and low ARS cell groups communicated *via* the GDF (GDF15-TGFBR2) and EGF (AREG-EGFR) pathways.

most LUAD patients as well as a poor cancer prognosis. With the development of surgical and drug-assisted therapies, novel and effective diagnostic methods for heterogeneous LUAD patients should be created to improve the survival outcomes of LUAD.

In this study, we focused on developing, establishing, and verifying the aneuploidy-related gene signature by integrating bulk and scRNA-seq data to thoroughly analyze the role of aneuploidy in the diagnosis and treatment of LUAD. The predictive model utilizing a range of functional genes has emerged as a focal point in directing the prognosis of tumors. *Wu et al. (2022)* constructed a prognostic model based on necroptosis-related genes and the area of the ROC curve of this model in the training cohort was 0.73, 0.7, and 0.67, respectively. In addition, (*Guo & Liu, 2022*) obtained the Transient receptor potential channel (TRP) gene set based on the MSigDB database and constructed a related risk model based on TRP. To assess the validity of the model, the results showed that the areas of the ROC curves in the training cohort at 1, 3 and 5 years were 0.747, 0.663 and 0.654, respectively (*Guo & Liu, 2022*). In this study, we constructed an ARS-related prognostic model in which the AUC values for the training set were 0.68, 0.65, and 0.67, respectively. In this study, we constructed an ARS-related prognostic model in which the AUC values were 0.68, 0.65, and 0.67 for the TCGA cohort and 0.77, 0.74, and 0.75 for the GSE30219 validation cohort. This result suggests that the predictive power of our constructed model is also reliable.

Through differential gene analysis, COX regression, Pearson analysis and Lasso method, five genes (PLK1, ANLN, HMMR, CKS1B and UBE2S) were screened to construct an ARS model. We found that patients in high ARS group displayed shorter OS time, which was in agreement with a previous research (*Zhang & Zhao, 2023*). At the same time, the risk model was validated and proven to be an independent indicator for the prognosis of LUAD. These five genes were all overexpressed risk factors in tumor tissues in comparison with para-cancerous tissues. Previous researches on these genes mainly have their focuses on the potential to serve as diagnostic biomarkers and the mechanisms of action. Polo-like kinase 1 (PLK1), an evolutionarily conserved Ser/Thr kinase, has the function to regulate cell cycle, and the imbalance of PLK1 could lead to chromosomal instability and aneuploidy (*Gheghiani et al., 2021*). PLK1 is high-expressed in various cancers and is related to unfavorable prognosis (*Galusic et al., 2020*). Increasing evidence suggested that PLK1 regulates several critical TFs in cancers and facilitates cell proliferation, metastasis, epithelial mesenchymal transition (*Moore, Gheghiani & Fu, 2023*; *Iliaki, Beyaert & Afonina, 2021*). These findings indicated that PLK1 may be a promising target for cancer treatment. Currently, many PLK1 inhibitors have been developed for disease treatments. An intriguing observation in our research was that patients in high ARS group were more sensitive to PLK1 inhibitors (BI-2536 and GW843682X). This finding further supported the potential of PLK1 to serve as a diagnostic and therapeutic marker for LUAD. ANLN is an actin-binding protein that is often overexpressed to cause a poor prognosis (*Long et al., 2018*) and metastasis (*Xu et al., 2019*) in LUAD, but it could be suppressed by miR-30a-5p (*Deng et al., 2021*). Hence, ANLN has been frequently selected as a hub gene to build riskscore signature to evaluate the prognosis in LUAD (*Luo et al.,*

*2020*; *Zhang & Zhao, 2023*). This suggested that these two genes could promote metastasis and development of LUAD cancer cells.

Hyaluronan-mediated motility receptor (HMMR) plays an important role in cancer progression. For example, overexpressed HMMR in mouse mammary gland epithelium increases the occurrence of BRCA1-mutant tumors, which is directly related to hereditary breast cancer (*Mateo et al., 2022*), while inhibition on HMMR expression could abrogate peritoneal metastasis of gastric cancer cells (*Yang et al., 2022*). A newly published pan-cancer analysis supported the latent capacity of HMMR as a diagnostic and prognostic indicator of tumor, because HMMR is widely high-expressed in massive tumor cells (*Shang et al., 2022*). A holistic pan-cancer analysis proved that CKS1B is a cancer-promoting factor (*Jia et al., 2021*). In addition, *Wang et al. (2021)* first characterized CKS1B as a potential prognostic marker in LUAD, and this gene was also discovered in our present work. Another pan-cancer analysis on UBE2S (*Bao et al., 2022*) revealed that UBE2S expression is negatively correlated with the prognosis of different human cancers and is noticeably positive related to MDSC and T helper 2 cell (Th2) subsets in most tumors, indicating that UBE2S is an immune-carcinogenic marker. This work highlighted the potential of using UBE2S in immunotherapy for LUAD patients.

To better understand the transcriptional regulatory networks across intracellular cancer cells, we predicted the TF-target regulating networks between high and low ARS cell groups. Different regulons in the two risk cell groups were discovered. In high ARS cell group, regulons CREB3L1 and CEBPD were enriched. Abnormal CREB3L1 expression results in growth, metastasis or migration of malignant tumors (*Wang et al., 2021*; *Mellor et al., 2022*). CREB3L1 could facilitate the above biological processes by reshaping the tumor immune microenvironment (*Pan et al., 2022*; *Song et al., 2022*). Interestingly, the potential of CREB3L1 to predict immunotherapeutic efficacy is also supported by a previous pan-cancer analysis (*Lin et al., 2022*). CEBPD is one of the hub genes for triple-negative breast cancer prognosis (*Kim et al., 2019*). We found that TFs BCLAF1 and UQCRB with over 900 downstream target genes were enriched in low ARS cell group. Studies also confirmed that during tumor development, BCLAF1 acts either as a tumor promoter (*Mou et al., 2020*) or a suppressor (*Li et al., 2018*) dependent on the cellular environment and category of tumor (*Yu et al., 2022*). Some studies uncovered that UQCRB is involved in tumorigenesis by recognizing its genetic mutations or relevant microRNAs in pancreatic ductal adenocarcinoma (*Harada et al., 2009*) or colorectal cancer (*Hong et al., 2020*). To the best of our knowledge, these four TFs were first described in LUAD cancer cells, but their mechanism underlying LUAD development required more in-depth investigation.

Aneuploidy is related to lower immune infiltration in many types of tumors (*Davoli et al., 2017*). Consistent with previous findings, high ARS group showed a low degree of immune infiltration, as shown by the results from ssGSEA and CIBERSORT analysis. Subsequent scRNA-seq data concerning CellChat analysis uncovered that high ARS cell group received and sent communication signals *via* SPP1(ITGA4-ITGB1) and MK(MDK-NCl) signaling pathways. Intriguingly, these two pathways help shape a tumor immune barrier (*Liu et al., 2023a*) to develop an immunosuppressive environment in tumor (*Yu*

*et al., 2023*). This finding may explain the apparently poor prognosis and less active response of LUAD patients to immunotherapy at molecular level.

Despite these interesting findings, there are also some limitations to this study. Firstly, the expression of the aneuploidy-related genes should be detected at protein level. In addition, the communication pathways between cancer and immune cells were obtained *via* bioinformatics algorithms and further verification is needed. Moreover, the effect of the prognostic genes in LUAD is required to be confirmed by *in vivo* validation. Finally, as all the data were obtained based on public databases, future work is encouraged to use a larger sample size to confirm the accuracy of the present findings.

## CONCLUSIONS

To conclude, we analyzed the bulk and scRNA-seq data related to aneuploidy in LUAD. An independent ARS signature for predicting LUAD patient's prognosis was established. The viability of the ARS model genes to predict the prognosis and progression of LUAD was tested by performing a series of cellular experiments, and the key ARS-related genes contributing to the malignant phenotypes of LUAD were mined. Our study provided a new direction for the biomarker research in LUAD, and confirmed that the ARS model can help improve the current personalized treatment for LUAD patients, especially in the selection of chemotherapeutic agents and immunotherapy strategies.

## ABBREVIATIONS

| | |
|---|---|
| **ARS** | Aneuploidy related riskscore |
| **AUC** | Area under ROC curve |
| **CR** | Complete response |
| **CSI** | Connection specific indicators |
| **DEGs** | Differentially expressed genes |
| **FDR** | False discovery rate |
| **GEO** | Gene Expression Omnibus |
| **GO** | Gene Ontology |
| **Lasso** | Least Absolute Shrinkage and Selection Operator |
| **IC50** | Half maximal inhibitory concentration |
| **KEGG** | Kyoto Encyclopedia of Genes and Genomes |
| **LUAD** | Lung adenocarcinoma |
| **MDSC** | Myeloid-derived suppressor cell |
| **NSCLC** | Non-small cell lung cancer |
| **OS** | Overall survival |
| **PCA** | Principal Component Analysis |
| **PD** | Progressive disease |
| **PR** | Partial response |
| **ROC** | Receiver operator characteristic |
| **sc-RNA seq** | Single-cell RNA sequencing |
| **SD** | Stable disease |

| ssGSEA | Single-sample gene set enrichment analysis |
| TCGA | The Cancer Genome Atlas |
| TFs | Transcription factors |
| t-SNE | t-distributed Stochastic Neighbor Embedding |

### Funding
The authors received no funding for this work.

### Competing Interests
The authors declare that they have no competing interests.

### Author Contributions
- Xiaobin Wang conceived and designed the experiments, prepared figures and/or tables, and approved the final draft.
- Jiakuan Chen conceived and designed the experiments, authored or reviewed drafts of the article, and approved the final draft.
- Chaofan Li conceived and designed the experiments, prepared figures and/or tables, and approved the final draft.
- Yufei Liu conceived and designed the experiments, authored or reviewed drafts of the article, and approved the final draft.
- Shiqun Chen performed the experiments, prepared figures and/or tables, and approved the final draft.
- Feng Lv performed the experiments, authored or reviewed drafts of the article, and approved the final draft.
- Ke Lan performed the experiments, prepared figures and/or tables, and approved the final draft.
- Wei He performed the experiments, authored or reviewed drafts of the article, and approved the final draft.
- Hongsheng Zhu analyzed the data, prepared figures and/or tables, and approved the final draft.
- Liang Xu analyzed the data, authored or reviewed drafts of the article, and approved the final draft.
- Kaiyuan Ma analyzed the data, prepared figures and/or tables, and approved the final draft.
- Haihua Guo analyzed the data, authored or reviewed drafts of the article, and approved the final draft.

### Data Availability
The data used in this study is available at NCBI GEO: GSE31210, GSE30219, GSE203360.

The raw data is available at GitHub and Zenodo:
- https://GitHub.com/1Guohh/Raw-data.git
- 1Guohh. (2024). 1Guohh/Raw-data: First release of my raw data (v.1.1.0). Zenodo. https://doi.org/10.5281/zenodo.10485770.

## Supplemental Information

Supplemental information for this article can be found online at http://dx.doi.org/10.7717/peerj.17545#supplemental-information.

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
