# Peer review of "Integrated bulk and single-cell RNA sequencing identifies an aneuploidy-based gene signature to predict sensitivity of lung adenocarcinoma to traditional chemotherapy drugs and patients' prognosis"

_PeerJ, doi:10.7717/peerj.17545_

## Round 0.1 · original submission · Major Revisions

Although the three reviewers have all given relatively positive comments on your manuscript, they still put forward many comments that deserve revision and attention. Please modify the manuscript point-by-point according to the reviewers' comments.

**Language Note:** PeerJ staff have identified that the English language needs to be improved. When you prepare your next revision, please either (i) have a colleague who is proficient in English and familiar with the subject matter review your manuscript, or (ii) contact a professional editing service to review your manuscript. PeerJ can provide language editing services - you can contact us at [email protected] for pricing (be sure to provide your manuscript number and title). – PeerJ Staff

Reviewer 1 ·

Basic reporting

Wang et al. attempted to develop biomarkers that capture aneuploidy signature, aiming to predict prognosis and drug responses in LUAD cancer. The authors used bulk RNA sequencing to separate the LUAD cohort into low vs high aneuploidy scores based on this transcriptomic analysis. The authors showed that the low ARS subgroup exhibited prolonged survival. Using single-cell RNA sequencing, the authors attempted to distinguish core transcription factors and characterize cell-to-cell interactions, specifically between cancer cells and immune cells, between the two ARS subgroups. The authors then showed that the ARS score may help infer therapeutic options, for example, the preference for EGFR inhibitor and Roscovitine in the low ARS subgroup.

1) Please clarify the choice of statistical tests for all figures. P-values were provided in some figures, but there are not enough details to explain what statistical approaches were used to test these significances.
2) The manuscript requires proper English proofreading. There are many grammar errors, typos, and sentence structure issues.

Experimental design

Please see in section 3.

Validity of the findings

1) Figure 1 attempt to prove that differentially expressed genes are related to aneuploidy, but these analyses were not carefully explained to justify whether these hallmarks were selected unbiasedly or manually selected. Preferably, the readers may want to see that these hallmarks are enriched unbiasedly, showing higher enrichment scores than those unrelated to aneuploidy.
2) In Figure 2, the authors attempt to demonstrate that the 5-gene signature can predict survival. Although the KM plots separate well between the low vs high ARS subgroups, the authors may need to justify the ROC area under the curve of 0.6-0.7, which may be considered low in most studies.
3) In Figure 3, the authors attempt to compare the hazard ratios of riskScore developed from ARS score, vs age, gender, and stage. For all datasets, the RiskScore from ARS gave poorer hazard ratio than staging regardless of its significance difference. The authors may need to elaborate in results and discussion sections on what these mean. To most readers, they may expect that the developed riskScore should perform better than cancer staging, hence the need for this new biomarker panel.
4) For Figure 4, the authors are suggested to elaborate on the details of migration and invasion studies. Presumably, these results are from the transwell assay, but there are not enough details in the result or method sections. In panel 4H-I, the authors claimed that the knockdown of CKS1B reduced proliferation, with significant tests, although the overall growth rate may not be dramatic to justify its strong involvement in cancer progression.
5) Figure 5 attempts to demonstrate the distinct immune landscape between the low vs high ARS subgroups. In our experience, CIBERSORT could give drastically different predictions of immune cell fractions than other toolboxes, such as MCPcounter. The authors are recommended to compare the results with other toolbox in an attempt to validate their results. Even better, some validation of immune cell scoring using other methodologies, such as multiplex IF of archival specimens are recommended to further support this claim. The authors are also suggested to develop more specific figure panels in Figure 5, to support their claims. Currently, panels A and B are inadequate for the readers to follow.
6) In Figure 6, the authors tried to prove the merit of these ARS biomarkers as predictive biomarkers. However, we cannot evaluate the efficacy of these results, unless compared to gold standard biomarkers or even with cancer staging (which was shown by the authors themselves to perform better in predicting prognosis).
7) It is unclear how figure 7 supports their ARS signature. The authors did not show any differences between the low vs high ARS groups here.
8) For Figure 8A, there is no annotation of what the column is. It is unclear which cell population these regulon analysis were extracted from.

Additional comments

-

Reviewer 2 ·

Basic reporting

no comment

Experimental design

no comment

Validity of the findings

no comment

Additional comments

The manuscript entitled “Integrated bulk and single-cell RNA sequencing identifies a signature based on aneuploidy related genes to predict sensitivity to traditional chemotherapy drugs and prognosis in lung adenocarcinoma” This study clearly states the aims, background, and objectives, providing a comprehensive overview of the research focus. Authors firstly used bulk RNA analysis identified a 5 aneuploidy related genes signature to predicted prognosis, immunotherapy, immune status for LUAD. One of 5 aneuploidy related genes, CKS1B, was found higher expression in LUAD cells, and promoted cancer cell progression. scRNA analysis further used for validation. The method is reasonable and logical, and clear language expression. There are a few critical comments and suggestions that need to be addressed.
The methodology doesn’t seem to be a novel approach as similar paper was published in head and neck squamous cell carcinomas. And should cited the reference.
• Liu, Yu, et al. "Identification of aneuploidy-related gene signature to predict survival in head and neck squamous cell carcinomas." Aging (Albany NY) 15.22 (2023): 13100.
Conclusion of Abstract: the results should be summarized first.
The study could benefit from a brief discussion of potential limitations or challenges faced during the study, such as any biases introduced by the datasets or potential confounding factors.
The p-value should be lowercase and italicized
Some writing styles to note, such as cluster Profiler package, 2-ΔΔCT method, 1×103 cells/well,
The English language should be improved to ensure that an international audience can clearly understand your text. I suggest you have a colleague who is proficient in English and familiar with the subject matter review your manuscript, or contact a professional editing service.
For Figure 4F, the bar of each picture should be marked
**in all figure legend should be unified
Although the authors used a publicly available data set in this article, the data content itself was derived from the human body. I suggest that the author declare this at the end of the "Data source and pro-processing" section. For example: "All data used in this article are from publicly available databases, which are available for unrestricted access and analysis and require no additional ethical approval. Our download and analysis processes fit the rules."
The potential clinical impact should have been mentioned in the conclusion.
Overall, the paper should be carefully revision for further reviewed.

Reviewer 3 ·

Basic reporting

The subject of this study was to predict the sensitivity and prognosis of lung adenocarcinoma (LUAD) to chemotherapeutic agents based on aneuploidy-related gene characterization by single-cell RNA sequencing. The study first obtained RNA sequencing data of LUAD from public databases for differentially expressed genes in LUAD relative to controls. Multiple types of Cox regression analyses were performed on these genes to construct prognostic models, and the accuracy of the prognostic models in predicting the prognosis of LUAD was verified by cellular experiments. Subsequently, the immune mechanisms of LUAD regulation by prognostic models were revealed by drug sensitivity analysis and immune microenvironment analysis. Finally, the specific mechanisms affecting the immune response to LUAD were revealed by single-cell RNA sequencing. In conclusion, the idea of this study is poorly developed, but the following issues still need to be addressed before publication:
1. The abstract section of this study does not describe the research logic of this article very clearly, please make further adjustments to the methods section, especially to highlight the important role of single cell analysis techniques in this article.
2. What are the research tools included in this paper for the immune microenvironment? Please refine the abstract and preface sections, this should be one of the main focuses of the full paper.
3. The introductory section provides a good overview of the background of LUAD, including its contribution to global cancer mortality, as well as the therapeutic challenges. However, further in-depth discussion on why existing treatments are insufficient, and in particular where existing immunotherapies fall short, could be provided, allowing this paper to focus on the study of LUAD from a single-cell perspective.
4. Although the use of single-cell RNA sequencing (sc-RNA seq) to explore cancer cell heterogeneity in LUAD samples is mentioned in the last part of the preface, a brief mention can be made as to why this method is superior to others and how it can help to solve the problems identified in the study.
5. Ensure that transitions between paragraphs flow naturally and logically. For example, when moving from the context of LUAD to a discussion of alloploidy, the connection between them and why this focus of research is critical to understanding the treatment and diagnosis of LUAD can be more clearly stated.
6. Please briefly explain why this paper uses multiple methods to elucidate the immune landscape; does using both ssGSEA and CIBERSORT ensure that the analysis covers the vast majority of immune cells? If the results of the two conflict, which method will prevail? Suggest elucidation in the results or methods section.
7. Regarding the description of Figure 9, it is recommended that more information be provided on how to interpret the specific significance of these communication pathways for LUAD progression. For example, the role and importance of the SPP1 signaling pathway, the MK signaling pathway, and the GDF and ECF signaling pathways in LUAD need to be discussed in more detail. And, in the description of the results in this section, it is recommended that a summary statement be added.
8. The description of the five genes (PLK1, ANLN, HMMR, CKS1B and UBE2S) in the Discussion section is too lengthy and it is suggested to divide this section into two parts. That is, these genes can be divided into two categories according to the regulatory outcome, for example, one category is related to cancer metastasis and the other category is related to immunity.
9. This paper describes the results of cellular communication too briefly and does not clearly state the mechanism of action of ligand-receptor pairs on tumor progression and immunotherapy, which should be an important molecular mechanism, please provide additional description.
10. The limitations section of this paper is actually not fully described, and it is recommended to add relevant ideas about subsequent experiments conducted to validate the bioinformatics results.

Experimental design

no comment

Validity of the findings

no comment

Additional comments

no comment

---

## Round 0.2 · Minor Revisions

Please be sure to make modifications point by point according to the reviewer’s requirements.

Reviewer 1 ·

Basic reporting

For the justification of ROC area under the curve, the authors added comparison to GSE31210 and GSE30219. The 1,3, and 5 year OS from these datasets were shared, but a little more discussion on how well the prediction from the TCGA cohort, compared to these 2 datasets will help the readers to understand better. Simple statement like 'the prediction from TCGA is comparable to .. ', if that is what the authors mean to say.

Appreciate the authors improvement of Figure 2 , adding the nomogram and justification of why risk score improves the prediction with Supplementary figure 1. These new figures should be quite essential in justifying the importance of risk score, so the authors are advised to reconsidered if some of these new panels should be put in the main figure, or as supplemental figures.

For figure 3, I can noticed that the new section 'transwell assay' is available in method section now. Perhaps explicit details at the figure legends. While the migration and invasion study showed significant effects of siCKS1B, The claim in H&I, as stated 'CKS1B inhibition also significantly suppressed the viability of H1299 and A549 cell lines (Figures 3H-3I)' was still a little too strong, considering the small change in growth rate. Perhaps the authors can explain this small effect on proliferation, but more influence effect on migration and invasion in result and discussion to reduce confusion that some readers may have.

Figure figure 4 (old figure 5), the authors stated ". We will enhance the reader's understanding and confidence by showing more clearly the distribution of scores for each immune cell subtype and, where possible, visualizing specific immune cell populations using methods such as multiple IF. We believe that these improvements will make our conclusions more robust and easier for readers to understand."

I do not see the improvement that the authors have claimed to perform: 1) comparison to other method of immune cell predictions e.g.MCPcounter, 2) comparison with other techniques such as multiplex IF. Please clarify.

For Figure 5 (old figure 6), the authors stated "We plan to include comparisons of cancer staging as well as other established biomarkers in future studies". I believe such analyses are critical for the audiences to even consider the merit of their work. So, please reconsider doing this comparison as part of this work.

I appreciate the elaboration on figure 7 and 8, to demonstrate the cell-cell communication uniqueness between the low vs high ARS subgroups.

Experimental design

no further comment.

Validity of the findings

no further comment

Additional comments

I hope the authors could kindly address some of the concerns above to help the readers understand their paper better. I hope these comments are helpful.

Reviewer 2 ·

Basic reporting

no comment

Experimental design

no comment

Validity of the findings

no comment

Additional comments

In this study, the author systematically explored the relationship between the prognosis of non ploidy related genes in LUAD and immunotherapy, and established a risk assessment model for further in vitro validation. The revised version has made significant improvements, with appropriate statistics and complete logic. The results have been well explained, and I do not have any new comments

Reviewer 3 ·

Basic reporting

The author constructed an ARS biomarker model containing 5 genes that were abnormally expressed in LUAD cell lines, particularly high expression of CKS1B, which promoted the proliferation, migration, and invasion characteristics of LUAD cell lines. The overall survival period of the low ARS group is longer, the degree of inflammation infiltration is higher, and it is more likely to benefit from immunotherapy. The integration analysis based on the ARS model provides a potential direction for improving the diagnosis and treatment of lung adenocarcinoma (LUAD). After modification, it meets the publishing standards.

Experimental design

no comment

Validity of the findings

no comment

Additional comments

no comment

---

## Round 0.3 · accepted · Accept

From an editorial perspective, I believe the authors have adequately addressed the reviewers' concerns, and the manuscript does not need to be sent for further review. Here are the reasons:

1. Detailed Responses and Revisions
- The authors have provided detailed responses to each reviewer's comment and made corresponding revisions to the manuscript. For example, regarding the discussion of the ROC curve, the authors added comparisons with GSE31210 and GSE30219 datasets and discussed this in the manuscript.
- For the improvement of Figure 2, the authors added a nomogram and explained how the risk score improves prediction accuracy. They also provided a detailed explanation of the nomogram in the main text.
- In response to comments on Figure 3, the authors revised the description of the effect of CKS1B inhibition on cell viability to avoid overstatement and explained the significant effects on migration and invasion.
2. Clear Explanations and Justifications:
- For the improvements to Figure 4, the authors explained their choice of ssGSEA and CIBERSORT methods and discussed the complexities and resource requirements that would come with introducing additional techniques such as multiplex immunofluorescence.
- Regarding Figure 5, the authors indicated that they plan to include comparisons of cancer staging and other known biomarkers in future studies but have discussed and evaluated these aspects in the current manuscript.
3. Positive Feedback and Acknowledgment:
- The authors not only provided detailed responses to each reviewer's comment but also expressed gratitude for the constructive feedback, demonstrating a positive and collaborative attitude.
- They emphasized their appreciation for the suggestions and showed commitment to improving and refining their work further.

Based on my careful review of each response to the reviewers' comments, I conclude that the authors have addressed all major concerns adequately. Additionally, Reviewer 2 and Reviewer 3 did not raise new issues but acknowledged the previous revised version. Therefore, I believe the manuscript has reached a publishable standard and does not require further review.